# geneDRAGNN - gene Disease PRioritizAtion using Graph Neural Networks

**Anonymous Authors** [1]

## Abstract

Most human diseases exhibit a complex genetic etiology impacted by many genes and proteins in a large network of interactions. The process of evaluating gene-disease association through in-vivo experiments is both time-consuming and expensive. Thus, network-based computational methods capable of modeling the complex interplay between molecular components can lead to more targeted evaluation. In this paper, we propose and validate geneDRAGNN: a general data processing and machine learning methodology for exploiting information about gene-gene interaction networks for predicting gene-disease association. We demonstrate that information about the gene-gene interaction network can significantly improve the performance of gene-disease association prediction models. We apply this methodology to predicting gene-disease association for lung adenocarcinoma, a histological subtype of lung cancer, and perform genomic analysis on genes flagged as potential associations.

## 1. Introduction

With the widespread adoption of direct-to-consumer gene sequencing and increased interest in precision medicine, uncovering the genetic mechanisms underlying diseases enables new avenues for disease prevention and treatment. Currently, linkage analysis and genome-wide association studies (GWAS) are the most widespread approaches to uncovering gene-disease associations. However, these approaches are limited by costly and time-consuming statistical analysis and validation of identified biomarkers due to a high rate of false positives (1). Moreover, these techniques are often focused on broad genotype-phenotype associations and fail to capture the complex molecular interactions that constitute broader biological systems.

While there exist many biological networks such as gene

regulatory networks and metabolic interaction networks, protein-protein interaction (PPI) networks are the most extensively used for predicting gene-disease associations. As proteins are encoded by genes, genetic mutations often also lead to disease as they can lead to protein abnormalities. This indicates that an enriched gene-gene interaction (GGI) network with additional domain knowledge, such as gene ontology, expression and functional features, could build a more complete and robust molecular environment. Graph Neural Networks (GNNs) can leverage these networks to predict disease associated genes.

GNNs are neural networks which take node data and a graph as input. In a general GNN, an embedding $h_v$ is computed for each node $v$ using its features and features of its neighbours. Using that embedding, the output $o_v$ is computed for each node. There is a local transition function, $f$, that aggregates information from neighbouring nodes and updates each node's embedding. After some predetermined number of updates, the embeddings are passed through a global output function, $g$, that produces the final prediction $o_v$.

$$h_v = f(\boldsymbol{x}_v, \boldsymbol{x}_{nb[v]}, \boldsymbol{x}_{co[v]}, \boldsymbol{h}_{nb[v]}), \qquad (1)$$

$$o_v = g(\boldsymbol{h}_v, \boldsymbol{x}_v) \qquad (2)$$

The functions $f$ and $g$ are learned through, for example, feed-forward neural networks. In this study, we propose a new framework, geneDRAGNN, for predicting gene-disease associations. geneDRAGNN aims to leverage PPIs while also incorporating other biological information. Specifically, gene ontology, tissue- and cell- specific gene expression, and mutation rate are used to enhance and annotate the network. To successfully model this data we employ GNNs.

The disease of interest is lung adenocarcinoma (LUAD) – a histological subtype of lung cancer. Lung cancer has the highest incidence (approx. 1.8 million new cases yearly) of all cancers, largest number of deaths globally (approx. 1 million yearly), and is the most frequent malignancy (2). The effectiveness of conventional treatments remains limited and the results of diagnosis indicate a poor prognostic outcome (3). Therefore, the discovery of novel genes which are associated with the occurrence and progression of LUAD is essential to identifying future treatments and prevention methods. In our research we make the following contributions:

---

*Equal contribution [1]Anonymous Department. Correspondence to: Anonymous Authors <anonymous.authors@mail.com>.

*Proceedings of the 38th International Conference on Machine Learning*, PMLR 139, 2021. Copyright 2021 by the author(s).

- We develop a general data collection and machine learning methodology geneDRAGNN for exploiting information about the GGI network to predict gene-disease associations.

- We apply our methodology to gene-disease association prediction for lung adenocarcinoma, and demonstrate that graph-based machine learning methods like graph neural networks (GNNs) outperform traditional machine learning methods which don't use network information.

- Using our GNN models, we identify a set of genes which are enriched for association with LUAD and conduct a simple genomic analysis to qualitatively assess the performance of the model.

In this paper, we build upon previous work in bioinformatics and graph-based machine learning, and apply it to the study of classifying gene association to LUAD. While there exists research on applying machine learning to predicting genes associated with cancer, to our knowledge, this is the first exploration of using graph-based machine learning to predict genes associated with a specific histological subtype of cancer.

## 2. Related Works

Machine learning methods have been applied with some success to the problem of gene-disease association prediction. In (4), the authors explore an ensemble machine learning approach for predicting gene-disease association using lexical, syntactic, and semantic features extracted from literature using Word2Vec. (5) employs a similar approach, combining text mining of the vast biomedical literature with machine learning to develop RENET: a deep learning approach which considers the correlation between the sentences in an article to extract gene-disease associations. (6) uses neural networks to predict gene-disease association using biological features, gene sequence length, the entropy of an amino acid sequence, the discrete wavelet transform of gene sequences, and topological features from gene interactions. An overview and comparison of network based techniques for gene-disease association classification for various cancers can be seen in (7).

In (8), the authors exploited the unique properties of graph neural networks to propose a network-based deep learning approach to prioritizing autism genes in the Human Molecular Interaction Network (HMIN). Their framework, Prioritization of Autism-genes using Network-based Deep-learning Approach (PANDA), estimates and ranks the probability of autism association for every gene in the network. It achieved a classification accuracy of 89%, outperforming other common machine learning algorithms, and identified genes that were found to be significantly enriched for autism association.

Since the publication of (8), more elaborate GNN architectures have been proposed with improved performance. This includes Graph Convolutional Networks (9), Graph Attention Network (10), Message-Passing Networks (11), *node2vec* (12), Graph SAmple and aggreGatE (Graph-SAGE) (13), Chebyshev Graph Neural Network (ChebNet) (14), and Topology Adaptive Graph Convolutional Network (TAGCN) (15).

## 3. Datasets

geneDRAGNN takes a comprehensive knowledge-based approach to gene-disease association by aggregating heterogeneous biological systems and gene functional data. Specifically, data used in this study are classified into three categories: (1) gene ontology and functional annotation, (2) GGI network, and (3) gene-disease associations. The GGI network (2) is annotated with gene-ontology (GO) terms and tissue- and cell-specific expression levels (1). We train our models on the annotated GGI network using known gene-disease associations (3) as positive labels.

### 3.1. Gene Ontology and Functional Annotation

In the geneDRAGNN framework, gene properties are provided to the graph learning models as node features. We use two primary datasets for gene features, the Human Protein Atlas and the National Institute of Health: Genomic Data Commons.

**Human Protein Atlas** The Human Protein Atlas (HPA) combines ontological, expression and immunohistochemistry-based data to create comprehensive proteomic and transcriptomic profiles of human tissue. The database was queried for genes that are expressed in lung tissue (n=15021). These datasets include common gene identifiers, ontological data, expression data, and additional features such as pathology prognostics and protein class.

The ontological data is organized into the following categories: Subcellular Location (n=35), Biological Process (n=224), and Molecular Function (n=123). HPA data also includes tissue and cell-specific expression data collected from in-house RNA samples of 52 frozen tissues, the Genotype-Tissue Expression consortium dataset (GTEx), and FAN-TOM5 consortium data. The latter two datasets were produced using deep mRNA sequencing and all data was normalized.

**National Institute of Health Genomic Data Commons** In addition to the gene ontology features, we used the National Cancer Institute: Genomic Data Commons (GDC) to

obtain frequency of mutation data, such as simple somatic mutations and copy number alteration gain and loss, from the TCGA-LUAD patient cohort (n=567) (16). The TGCA-LUAD target cohort comprised 19,123 out of 21,092 genes which had at least 1 simple somatic mutation.

## 3.2. Protein-Protein Interaction Network

The STRING dataset is a composite PPI network containing multiple categories of protein interactions saved in an edgelist format. Edges represent interaction between two proteins and evidence for each edge is derived from a broad range of curated and literature based evidence. (17). Specifically, the STRING PPI network includes protein interactions from experimental evidence, genomic context, curated databases, and homology. As such the STRING PPI is information rich and contains 11,938,498 edges.

## 3.3. Gene-Disease Associations

DisGeNet is a widely used discovery platform for gene-disease associations (GDA). The platform consolidates data from curated repositories, GWAS catalogs, animal models, and the scientific literature to rank potential GDAs. Factors such as the number of publications, type of publication, and publication source are used to create a weighted association score. In addition, DisGeNet also provides an evidence index (EI). The EI is derived from the BEFREE and Psy-GeNET datasets and measures the degree of contradictory sources for a GDA. In this study, we use the DisGeNet platform to gather known LUAD-gene associations.

## 4. Methodology

### 4.1. Gene Ontology (Node Features) Processing

Gene ontology terms, protein class, and disease involvement features were one-hot encoded. This resulted in a sparse matrix encoding of our GO features on which we applied singular value decomposition (SVD) to reduce dimensionality. Multiple configurations of the SVD embeddings were tested on a random forest classifier and the embedding yielding the highest accuracy (100-component SVD) was used for the remainder of our modeling experiments.

### 4.2. Tissue-Specific Gene-Gene Interaction Network

In this study, we present the Lung Specific Functional Gene Interaction Network (LS-fGIN), a GGI network that prioritizes genes in the lungs. LS-fGIN is a subset of the STRING PPI such that each interaction is represented by one orientation resulting in an undirected graph. Nodes are mapped from Ensembl protein identifiers to their corresponding Ensembl gene identifiers using the STRING protein aliases dictionary. The conversion is a many-to-one mapping where multiple protein identifiers map to a single gene identifier. Finally, LS-fGIN is subset to include only genes expressed in the lung.

## 4.3. Gene-Disease Association Score Processing

Using the Disease Gene Network (DisGeNet) portal, we collected the Gene-Disease Association (GDA) scores of all genes associated with LUAD. The corresponding Evidence Indices, which indicate the percentage of support for the GDA, were also extracted. To increase the confidence in our GDA labels, we applied a GDA score threshold of 0.02 and Evidence Index threshold of 0.7 to ensure the candidate genes had sufficient evidence to support the association while maintaining a large sample size.

As defined in (18), we used the Selected Completely At Random (SCAR) assumption to develop negative labels. Random samples were collected from the unknown labels of the Disease Gene Network to create a set of negative labels.This methodology assumes that most genes are not associated with LUAD so that the true label of the sampled genes is negative. To verify this assumption and account for sampling bias, 100 samples of negative labels were created and results over the 100 trials were averaged to determine a final score.

## 4.4. Models

### 4.4.1. BASELINE NODE-ONLY MODELS

To evaluate the impact that the network information has on the performance of our models, a selection of baseline models were trained to establish a benchmark for the graph-based models. These models use only features about the genes (nodes) themselves as described in Section 3.1 and Section 4.1. The models tested include Multilayer Perceptrons (MLP), Random Forests (RF), K-Nearest Neighbours (KNN), and Support Vector Machines (SVM).

### 4.4.2. GRAPH-BASED MODELS

*node2vec*

*node2vec* is a method of graph representation learning which learns a low-dimensional vector representation of nodes, edges, and subgraphs through a biased random walk procedure (12) (19). We applied *node2vec* to LS-fGIN to generate neighbourhood-based network features for each node. We set the dimension of the generated embeddings to 128, the number of walks to 10, the walk length to 80, the window size to 10, and the number of epochs to 10. Lastly, we concatenate the features generated by *node2vec* with the node features described in Section 4.1 and trained MLP, RF, and SVM models using the same process as our baseline.

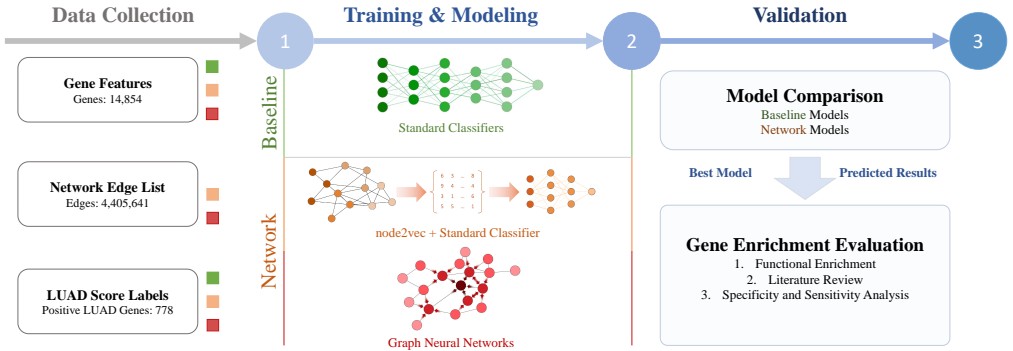

*Figure 1.* geneDRAGNN consists of three main components. Data is collected on gene ontology and functional annotation features, the GGI network, and GDA scores. These features are used to train graph-based machine learning models as well as benchmark models which perform gene-disease association prediction. A model is chosen based on performance to predict gene association with LUAD of unlabeled genes and the results are validated through biological enrichment analysis.

GRAPH NEURAL NETWORK

In our experiments, we used a common architecture for our GNN as depicted in Figure 2, and iterated with different graph convolution layers.

The GNN architectures we tested include Graph Convolutional Networks (GCN) (9), Simple Graph Convolutional Networks (SGC) (20), GraphSAGE (13), Graph Attention Networks (GAT) (10), ChebNets (14), and Topology Adaptive Graph Convolutional Networks (TAGCN) (15). ChebNets, GCNs, and GraphSAGE are examples of successful spectral approaches towards graph convolution operators which propagate information across nodes. In spectral methods, the graph signal is transformed to the spectral domain, processed, then transformed back using graph fourier transforms. ChebNet employs approximations of the Chebyshev polynomial as learnable filters. GCNs build on ChebNets and simplify the convolution operation. SGCs further simplify GCNs by removing non-linearities and collapsing weight matrices between consecutive layers, demonstrating in the process that the low-pass filter behavior of GNNs is the essential component for their success. The GraphSAGE framework generates embeddings by sampling from a node's neighbourhood and performing some form of aggregation through an LSTM, pooling, or averaging. GATs are an example of attention-based spatial approaches and incorporate an attention mechanism into the propagation step (19).

The choice of hyperparameters is motivated by the network properties described in Section 5.1. In particular, the diameter of LS-fGIN motivates a shallow network of at most 4 graph convolution layers. The large size of the graph also places restrictions on computationally tractable architec-

tures. For example, GNN architectures which process edge features were not computational tractable on the full GGI network.

### 4.5. Model Evaluation

A consistent methodology is used in the development and training of each model to enable an unbiased comparison. Each model is trained for 100 different trials each corresponding to a label set as described in Section 4.3. The use of multiple trials with different sets of negative labels provides confidence that our models' performance generalizes to the full set of genes. On each trial, independent training (70%), validation (10%), and test (20%) sets are created. At the end of each trial, a common set of metrics are calculated on the training and test sets including: accuracy, precision, recall, and f1-score. Recall is the more important metric for the purposes of this paper since false negatives are more costly than false positives. The comparison between models is made on the basis of the average performance of their 100 trials on the test sets as well as the stability of this level of performance. Each trial consisted of a balanced dataset of 1556 genes.

## 5. Results

### 5.1. Network Properties

Table 1 describes the fundamental network properties of LS-fGIN. LS-fGIN is highly connected, containing 1 strongly connected component and node degrees ranging from 1 to 6282 with a mean degree of 593.193. Interestingly, LS-fGIN does not follow a power law distribution and instead displays qualities of a Erdös-Rényi network. The Erdös-Rényi model describes random graphs and is characterized by a

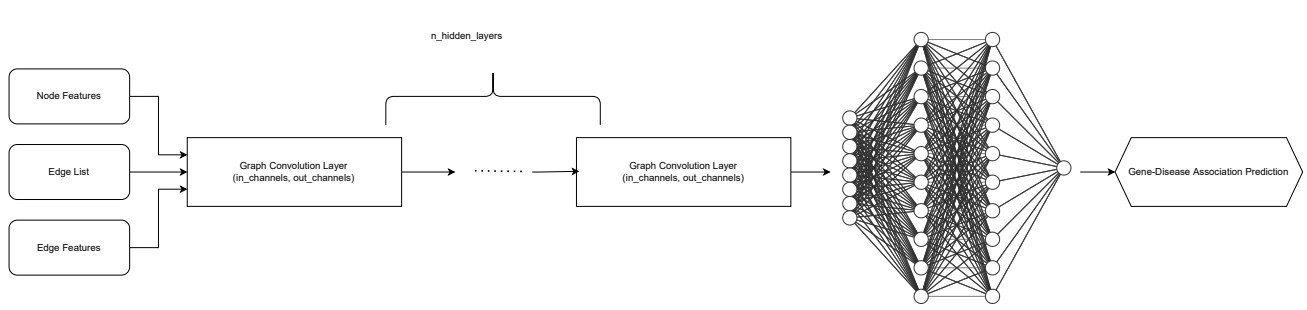

*Figure 2.* The general architecture of our graph neural networks.

Poissonian degree distribution for large graphs. Moreover, the probability of connections between two pairs of vertices is approximately the same for each pair. Erdös-Rényi graphs tend to have a larger central component containing the majority of connections, as seen in LS-fGIN (21).

*Table 1.* Properties of the gene-gene interaction network

| Property | |
|---|---|
| Number of nodes | 14854 |
| Number of edges | 4405641 |
| Number of connected components | 1 |
| Network diameter | 4 |
| Network density | 0.0399 |
| Clustering coefficient | 0.1998 |
| Average node degree | 593.193 |
| Average shortest path length | 1.998 |

## 5.2. Classification Results

### 5.2.1. BASELINE MODELS

The goal of the baseline models is to establish a benchmark for the network based methods and measure the added predictiveness and value of network information. Gene ontology and expression data were found to be predictive of gene-disease association, with baseline node-only models achieving roughly 69% accuracy. MLPs, Random Forests, K-Nearest Neighbours, and Support Vector Machines classifiers were used and achieved accuracies of 69.7%, 71.2%, 64.5%, and 69.1%, respectively.

### 5.2.2. GRAPH-BASED MODELS

Section 5.2.2 shows the model performances. In general models using LS-fGIN consistently outperformed the baseline models trained exclusively on gene-ontology features. Graph-based models achieved a maximum accuracy, precision, and recall of 78%, 80%, and 75% respectively. The model which achieved the highest average accuracy was the Support Vector Machine trained on gene-ontology features

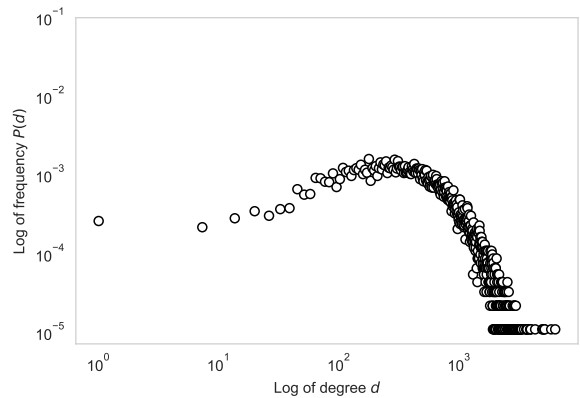

*Figure 3.* The log-scale degree distribution of LS-fGIN. The distribution does not follow a power law distribution as expected but instead approximates the degree distribution of a Erdos Renyi network (22)

and *node2vec* embeddings of LS-fGIN with an average accuracy of 78.0%. The highest performing GNNs achieved an average accuracy of roughly 75%. SGConv achieved an average accuracy of 74.3% with a standard deviation of 2.7%, and an average precision of 74.6%. The SGConv model had the highest positive recall of all GNN models, averaging 75.0%.

## 5.3. Validation of prioritized LUAD genes

We use the DAVID functional annotation tool to perform enrichment analysis on the top-ranking genes as predicted by the SGConv GNN model (chosen on the basis of its high recall). Specifically, we looked for enriched GO terms and KEGG pathways in the highest-ranked decile and top ten unlabeled genes classified by our chosen model. We perform a similar analysis on the top ten genes classified by our MLP model to qualitatively assess functional differences between classifiers in the top-ranked genes. **??** shows the top ten enriched GO terms. In addition to functional

*Table 2.* The performance achieved by each model which was tested. The baseline models use only node features–features about the genes themselves. The graph-based models use features about the gene-gene interaction network (either via *node2vec* embeddings or a GNN processing the graph directly). The metrics are averaged over multiple trials. Some models were evaluated on fewer than the full 100 trials due time and computational constraints. The more promising models were evaluated on the full 100 trials.

| Model | Features Used | Accuracy | Positive Recall | Positive Precision | F1-Score | # of Trials |
|---|---|---|---|---|---|---|
| *Baseline Models* | | | | | | |
| Random Forest | Node features | 0.707 | 0.728 | 0.699 | 0.707 | 100 |
| MLP | Node features | 0.699 | 0.668 | 0.713623 | 0.699 | 100 |
| K-Nearest Neighbours | Node features | 0.645 | 0.45 | 0.737 | 0.630 | 100 |
| Support Vector Machine | Node features | 0.693 | 0.506 | 0.809 | 0.681 | 100 |
| *Graph-based Models* | | | | | | |
| Random Forest | Node Features, *node2vec* Network Features | 0.766 | 0.705 | 0.802 | 0.765 | 100 |
| K-Nearest Neighbours | Node Features, *node2vec* Network Features | 0.645 | 0.751 | 0.620 | 0.640 | 100 |
| Support Vector Machine | *node2vec* Network Features | 0.780 | 0.759 | 0.794 | 0.780 | 100 |
| MLP | Node Features, *node2vec* Network Features | 0.744 | 0.735 | 0.749 | 0.744 | 100 |
| MLP | *node2vec* Network Features | 0.731 | 0.736 | 0.730 | 0.730 | 100 |
| SGConv GNN | Node Features, Functional Graph | 0.743 | 0.750 | 0.746 | 0.741 | 100 |
| TAG | Node Features, Functional Graph | 0.749 | 0.706 | 0.778 | 0.747 | 10 |
| TAG | Node Features, *node2vec* Network Features, Functional Graph | 0.741 | 0.726 | 0.750 | 0.741 | 10 |
| ClusterGCN | Node Features, Functional Graph | 0.726 | 0.671 | 0.757 | 0.724 | 11 |
| GraphSAGE | Node Features, Functional Graph | 0.714 | 0.674 | 0.733 | 0.713 | 10 |

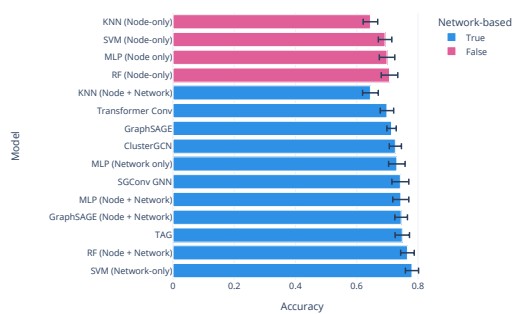

*Figure 4.* Average accuracies of the baseline and graph-based models over their trials. Error bars represent standard deviation of accuracy across trials and indicate the stability of each model at its average level of performance.

enrichment analysis, we also examined literature surrounding the top ten ranked unlabeled genes from both our SG-Conv model and MLP model. Literature was chosen using the DISEASES tool (23), which aggregates all literature on potential gene-disease associations. Overall, we found literature-based evidence for LUAD association in eight out of the top ten unlabeled genes classified by SGConv. Table X shows a selection of our top-ranked genes, a functional description of the gene, and literature-based evidence supporting a gene-disease association. Lastly, we sought to check the functional enrichment of benchmark gene sets from the Cancer Census (n=575) (16), DisGeNet labels with GDA < 0.01 (n=1755), and Cystic Fibrosis (n=2507) in our ranked deciles. We chose cystic fibrosis to test classification specificity as cystic fibrosis is a genetic disorder of the lung. Figure X shows the fraction of the respective gene sets in

each decile and highlights significantly enriched deciles. Using a one-tailed binomial test for significance we found that all three gene sets, Cancer Census, DisGeNet LUAD-gene associations with GDA < 0.01, and cystic fibrosis were enriched in the top decile.

## 6. Discussion

In this study we proposed and validated a general methodology for exploiting GGI network information for predicting gene-disease association using machine learning. The motivation behind this research is that most human diseases exhibit a complex genetic etiology impacted by many genes and proteins in a large network of interaction. Thus, leveraging information about this interaction network may be helpful in predicting associations between genes and diseases. We applied this methodology to the task of flagging genes associated with lung adenocarcinoma.

The GNN model achieving the highest recall was SGConv. For prioritization of LUAD genes, we use recall as our primary evaluation metric as the purpose of this study is to screen for novel gene-disease associations. Flagging genes using a high recall model has a purpose for further research and clinical action too, as identifying potential gene candidates can accelerate diagnostics turnaround and potentially drug discovery. Moreover, if the model can efficiently identify positive genes based on molecular characteristics, then it is efficiently trained to recognize similar characteristics in unlabeled genes.

Using SGConv, LS-fGIN nodes were scored on LUAD association and functional enrichment analysis was performed on the top ranked genes. The PI3K-AKT signaling pathway was found to be significantly enriched in the top-ranked

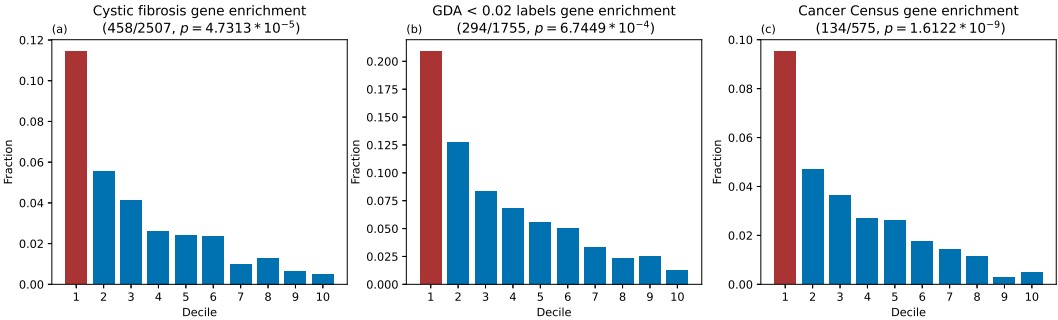

*Figure 5.* Distribution of benchmark gene sets among ranked deciles. (a) Cystic fibrosis associated genes from DisGeNet (b) lung adenocarcinoma associated genes with GDA score < 0.02 threshold (c) Cancer Gene Census. Significantly enriched deciles are highlighted in red.

decile with a p-value of $2.8E-14$. Activated by PI3K, AKT is a protein kinase that regulates fundamental cellular functions such as transcription, translation, proliferation, growth, and survival (24). Aberrant activation of the PI3K-AKT signaling pathway is often associated with tumor progression and resistance to cancer therapies (25). (26) has shown that the PI3K-AKT pathway is upregulated in NSCLCs, and is thus associated with LUAD.

Literature-based validation was also performed on the top 10 unlabeled genes and can be found in the Appendix A.2. Here we highlight FYN, a carcinogenic protein kinase with the highest prediction score. FYN plays an important role in the cell migration of lung carcinoma epithelial (A549) cells and is associated with the PI3K-AKT pathway (27). Overexpression of FYN reduces the migration and invasion capacities of A549 cells via down-regulating the PI3K/AKT pathway, thus effectively reducing tumorigenic capacity (28). (28) found that FYN levels were suppressed in LUAD tissue, allowing for AKT mediated tumorigenesis and ultimately supporting positive association with LUAD. This finding is significant since, in addition to supporting our functional enrichment analysis of the PI3K-AKT pathway, this finding also provides substantive and experimentally-verified evidence that geneDRAGNN is capable of identifying new, previously unlabeled, positively associated genes.

It is important that the findings made by geneDRAGNN are interpreted with regards to the methodology used for generating labels. In particular, while positive-label associations were based on experimentation in the literature, negative labels were obtained through random sampling. This is a common practice in similar bioinformatics machine learning research since negative association is very difficult or impossible to prove. For this reason, we ran multiple trials with different random samplings of negative labels as it ensures that the performance levels achieved are not due to random chance and the particular set of negative labels generated.

Computational limitations also restricted the size of possible models more generally. It is possible that larger models would have achieved better performance. In our case, the primary bottleneck was memory limitations caused by the large size of LS-fGIN which must be kept in memory during training. With more memory, weighted edges, or edge features, could be incorporated and potentially improve performance. Therefore, a promising direction for future research is to explore the potential benefits of using edge features and larger graph-based models.

Ultimately, geneDRAGNN successfully identified new genes associated with LUAD and has demonstrated the importance of taking an integrative approach to gene-disease association. Furthermore, we demonstrate that appropriate utilization of this information in machine learning methods can produce models which identify genes associated with a given disease with high accuracy, precision, and recall. In particular, different flavours of GNNs and network embedding methods like node2vec are effective methods for processing and utilizing network information for gene-disease association prediction.

## Data Availability Statement

This study uses open source data from the Human Protein Atlas, the National Institute of Health, the Disease Gene Network, and STRING. Our code, along with instructions for reproducing our results, can be found at `https://github.com/QMINDProjectX/geneDRAGNN`.

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

# A. Appendix

### A.1. Baseline Model Parameters

The MLPs used a simple architecture with 2 hidden layers of 128 units each and ReLU activation, implemented with PyTorch (29). The number of estimators of the random forests was set to 100. The K-Nearest Neighbors classifier used k=5 and the standard euclidean metric. The Support Vector Machine classifier used the Radial Basis Function kernel. Random Forests, K-Nearest Neighbours, and Support Vector Machine classifiers were implemented with Sci-Kit Learn (30).

### A.2. Graph-Based Model Parameters

Our implementations used the 'PyTorch Geometric' library (29). First, we converted LS-fGIN to a format readable by PyTorch Geometric by mapping our Ensembl gene identifiers to a 0-based indexing system and creating an edge list of shape `[2, n_edges]`. The edge features, which are not used by all our models, are represented by `[n_edges, n_edge_feats]` tensors. The Adam optimizer is used (31) with the cross-entropy loss and models are trained for 250 epochs. Throughout the training process, the model is evaluated on the training and validation sets, and model checkpoints are saved along the way. At the end of the 250 epochs, the model with the highest validation accuracy is restored and evaluated on the test set. This completes one trial.

| Rank | Gene | Gene functional description and Literature review |
|---|---|---|
| 1 | FYN | FYN encodes a tyrosine-protein kinase essential in cell motility and adhension and plays an important role in the PI3K/AKT pathway responsible for regulating the cell cycle (32). (28) demonstrated that overexpression of FYN accelerated cell apoptosis and reduced both angiogensis capacity and PI3K/AKT expression levels in lung carcinoma A549 cells. Conversely, FYN expression was shown to be correlated with LUAD prognosis as FYN expression levels were shown to be down-regulated in LUAD tissues and cells . |
| 2 | CDC42 | This gene encodes a member of the Rho subfamily of small GTP-binding proteins and plays a key role in cancer cell migration and metastasis. (32; 33) found decreased levels of StarD13, a surpressor of CDC42, in lung tumor tissue and A549 cells subsequently leading to increased CDC42 activation thus increasing formation of invadopodia, a unique hallmark of cancer, and matrix degradation. |
| 3 | PTPRC | This gene encodes a tyrosine-protein phosphatase required for T-cell activation. Upon T-cell activation, PTPRC recruits and dephosphorylates FYN which has been shown to be correlated with LUAD by (32). (34) directly supports PTPRC association with LUAD by demonstrating that PTPRC was a key gene in affecting the immune state of the tumor microenvironment and was ultimately correlated with a variety of tumor-infiltrating immune cells. |
| 4 | CREB1 | This gene encodes a phosphorylation-dependent transcription factor that stimulates transcription upon binding to DNA cAMP response element (CRE) (32). (35) used protein expression assays to understand the underlying mechanism of ferroptosis, a new form of regulated cell death associated with cancer, in LUAD. They found that CREB was highly expressed in LUAD and knockdown of CREB inhibited cell viability and growth by promoting apoptosis- and ferroptosis-like cell death. |
| 6 | ITGB1 | ITGB1 encodes the Integrin beta-1 subunit which when associated with Integrin alpha-3 provides a docking site for FAP, a serine protease involved in extracellular matrix degradation and tumor growth, at invadopodia plasma membranes. Hence ITGB1 may participate in formation of invadopodia, matrix degradation, and can promote cell invasion (32). Immune infiltration analysis revealed that the ITGB1-DT/ARNTL2 axis may effect the progression of LUAD and the immune microenvironment. ITGB1-DT/ARTL2 (36) |
| 9 | ISG15 | ISG15 acts as a cytokine, modulating immune responses, and can delay tumor cell growth by inhibiting tumor cell proliferation and angiogenesis. (37) found that high expression of ISG15 serves as a positive prognostic marker for long-term survival in LUAD patients. ISG15 has a broad network of protein targets, and (38) concludes that covalent ISG15 conjugation enhances the tumor-suppressive activity of the carboxyl terminus of Hsp70-interacting protein (CHIP), thereby showing an antitumor effect of Type 1 interferon. |
| 10 | LRRK2 | In an analysis of TCGA LUAD RNA-seq data, (39) identified that decreased LRRK2 expression is associated with LUAD. In (40), reduced LRRK2 expression was found to promote LUAD tumorigenesis and was associated with poor survival outcomes. This study also found overactivity in LRRK2 contributes to Parkinson's disease, which suggests pathological links between neurodegenerative disease and cancer are emerging. |