# OpenReview forum: "geneDRAGNN - gene Disease PRioritizAtion using Graph Neural Networks"
_uoft.ai/University_of_Toronto/2021/ProjectX — ProjectX2021_

### Official Review · Reviewer_pGYi · 2022-02-09
**geneDRAGNN - gene Disease PRioritizAtion using Graph Neural Networks**

**Rating:** 9
**Confidence:** 5

**Review:**

The authors implemented a GNN on PPI network, and used other biological features such as gene expression, gene-disease relationship as features to train the GNN to predict diseases. The authors did a thorough job benchmarking the results from GNN with other baseline methods. The results showed good improvement.

The described work is technically sound and close to the state of the art in the field.

The limitations I an think of are: additional important information such as genetic interactions, results from genome wide CRISPR screen, data from model organisms such yeast and fruit fly, can be incorporated into making the network or updating the network.

---

### Official Review · Reviewer_LmVB · 2022-02-10
**Very good**

**Rating:** 9
**Confidence:** 4

**Review:**

This paper addresses the problem of predicting gene-disease association by using gene-gene interaction networks. The authors propose to use graph-based neural networks (GNN) to leverage the existing knowledge that is a form of graph, which has not been done in prior work. The evaluation is done by comparing different classification models using the graph information against baselines that do not use the graph information on the lung adenocarcinoma (LUAD, a subtype of lung cancer) dataset. The experiment results show better performance when using graph information. The authors also provide additional analyses -- e.g., identifying important genes for LUAD, qualitative analysis,..

I enjoyed reading the paper. It is well-written and easy to follow. The paper clearly states background, motivation, problem, approach and its contribution to the field, addressing the gap in the previous work. The proposed methods are properly chosen, well-reflecting the characteristics of the data. If using the graph information is indeed new (not tried before as claimed in the paper), I strongly recommend publishing this paper in a good journal/conference venue after more revisions. The authors made their code available.

Below are my comments. They are mostly minor.

- the motivation is well explained-- although I'm not familiar with the bio/medical stuff.
- change the citation format (e.g., (1) => [1]) because it is confusing with the numbers with the same format in the paper. For example, the formula number and the numbering for three categories of the data in Section 3.
- the symbols in the formulas (1) and (2) should be explained either in the paragraph above and below, or by adding "where x is XXX, ..."
- In the contribution summaries: you developed a data collection methodology, too? <= My understanding is that the authors used existing datasets, so I don't understand how the authors use the term "data collection" in the paper. In Figure 1, there is also "data collection". The dataset section seems to describe existing datasets.
- Maybe specifying that the model in 4.4.2 is the proposed model would be more clear. <= later in the paper, I realized that the authors explored different architectures/algorithms to use graph information. This was not super clear in the first part of the paper, so I thought the authors proposed one model.
- stronger deep-learning based baselines not using the graph structures?
- evaluation description: Did the 100 trials use different splits of the same dataset? In other words, was the balanced dataset of 1556 genes split 100 different ways?
- for reproducibility, the model hyperparameters should be provided as well as whether hyper-parameter tuning was performed. If so, how. <= I saw some parameters are described in the appendix. It would be good to refer to the appendix in the model evaluation section as mentioning model parameters. But, I didn't find learning rate, batch size, etc. in the appendix, either.
- Table 2 needs to be mentioned in the result section.
- In the first paragraph of Sec 5.3, "??" needs to be fixed.
- Overall the later part of the paper seems to need more work to refine the writing, compared to the earlier part.

---

### Official Review · Reviewer_Jjox · 2022-02-15
**Solid results and well written paper**

**Rating:** 7
**Confidence:** 3

**Review:**

This paper presents geneDRAGNN, a graph neural network based approach used to predict gene-disease associations.  The proposed approach is specifically evaluated on a subtype of lung cancer.  The method leverages graph-based knowledge representations of various forms of protein-protein interactions and relationships as well as gene-disease interactions.  A wide range of methods are considered both using and not using the additional knowledge embodied in the graphs.  The results support the claim that the additional, graph-based information is valuable to predicting gene-disease relationships.  Beyond evaluating different machine learning approaches to determine the best performing method, the paper also explores and attempts to validated specific predictions made by their method, providing compelling evidence of the value of their approach.  No significant effort was spent to explore the performance differences of different learning methodologies built on top of the different representations, however this is not critical in my view.  The primary goal was to explore and demonstrate the usefulness of graph-based representations of knowledge and GNN-based processing in the prediction task.

Overall the paper is well written and the problem space well explored.  A solid contribution.

---

### Official Review · Reviewer_vx2Y · 2022-02-16
**This study sets up and tests the performance of different GNNs at predicting genes related to a specific disease (here, LUAD) based on gene features and those of the interactors in a PPI. The overall study is interesting, but the setup on the initial network and the "functional validation" have some serious weaknesses.**

**Rating:** 6
**Confidence:** 5

**Review:**

This study aims to benchmark different graph NNs for the task of relating genes to disease (in a very broad sense). The overall approach is to construct a PPI specific to the disease tissue of origin (here, lung), adorn the nodes with a series of features (general, and disease related) and then compare a baseline node-only model to one based on many different flavors of graph NNs. The problem is timely, and benchmarking can be helpful in this setting, given the many available approaches to graph NNs in the field. However, the "devil is in the details" and thus my main concern is in how the data is set up, and in how the evaluation is done.

1. The authors rely on HPA as their main source of "expression data" supplemented wish sources like GTEx, but as far as I can see they do not use LUAD expression data, and they do not use single cell RNA-seq data from LUAD or from healthy lung, both of which are available. As a result, their input is far noisier and less specific than it should be given the available data in the field. LUAD expression is different than healthy lung, and single cell expression is crucial to avoid confounding.
2. To the best of my understanding the authors use raw mutation frequencies. But LUAD in smokers has a very high mutation frequency with specific mutation signatures (due to tobacco) and hence, calling significant mutations is crucial. There are plenty of approaches for this it may be that the authors used this or pulled properly processed data from GDC. There is no way of knowing.
3. Their underlying PPI has properties (Erdos-Renyi) that make on seriously concerned. This is not typical of PPIs in biology, and I wonder if they applied proper quality filters. Could this explain why graph NNs perform worse than  some of their baselines?
4. The final "validation" should not be called validation. I appreciate the challenge in validating, I would call it at most "characterization". A top gene like PTPRC (aka CD45) is a pan-immune cell marker. Immune cell infiltration can absolutely affect LUAD survival, but I do not think this is what the authors were calling a "disease gene" (and if it is what they were looking for, they should have handled the model differently)
5. Computationally, the authors test many models, but we learn little about what may be the pitfalls of each.
6. I do not agree that false negatives are more costly than false positives. This may be the case for diagnostics (but this is really not the right framing for a dx problem), but for a biological/functional discovery or drug discovery, false positives are much "costlier" problem.
7. Please do not confuse protein protein interactions (a molecular/biochemical event) with gene-gene interactions (a genetic term).

---

### Decision · Program_Chairs · 2022-02-19

Winner